# Therapeutic Nanodiamonds Containing Icariin Ameliorate the Progression of Osteoarthritis in Rats

**DOI:** 10.3390/ijms242115977

**Published:** 2023-11-05

**Authors:** Ying Yu, Sang-Min Kim, Kyeongsoon Park, Hak Jun Kim, Jae Gyoon Kim, Sung Eun Kim

**Affiliations:** 1Department of Orthopedic Surgery and Nano-Based Disease Control Institute, Korea University Guro Hospital, Seoul 08308, Republic of Korea; yuying613358888@163.com (Y.Y.); heavystone75@gmail.com (S.-M.K.); dakjul@korea.ac.kr (H.J.K.); 2Department of Systems Biotechnology, Chung-Ang University, Anseong 17546, Republic of Korea; kspark1223@cau.ac.kr; 3Department of Orthopedic Surgery, Korea University Ansan Hospital, Korea University College of Medicine, Ansansi 15355, Republic of Korea

**Keywords:** icariin, tannic acid, nanodiamonds, cartilage degradation, inflammation, osteoarthritis

## Abstract

In present study, icariin (ICA)/tannic acid (TA)-nanodiamonds (NDs) were prepared as follows. ICA was anchored to ND surfaces with absorbed TA (ICA/TA-NDs) and we evaluated their in vitro anti-inflammatory effects on lipopolysaccharide (LPS)-activated macrophages and in vivo cartilage protective effects on a rat model of monosodium iodoacetate (MIA)-induced osteoarthritis (OA). The ICA/TA-NDs showed prolonged release of ICA from the NDs for up to 28 days in a sustained manner. ICA/TA-NDs inhibited the mRNA levels of pro-inflammatory elements, including matrix metalloproteinases-3 (MMP-3), cyclooxygenase-2 (COX-2), interleukin-6 (IL-6), and tumor necrosis factor-α (TNF-α), and increased the mRNA levels of anti-inflammatory factors (i.e., IL-4 and IL-10) in LPS-activated RAW 264.7 macrophages. Animal studies exhibited that intra-articular injection of ICA/TA-NDs notably suppressed levels of IL-6, MMP-3, and TNF-α and induced level of IL-10 in serum of MIA-induced OA rat models in a dose-dependent manner. Furthermore, these noticeable anti-inflammatory effects of ICA/TA-NDs remarkably contributed to the protection of the progression of MIA-induced OA and cartilage degradation, as exhibited by micro-computed tomography (micro-CT), gross findings, and histological investigations. Accordingly, in vitro and in vivo findings suggest that the prolonged ICA delivery of ICA/TA-NDs possesses an excellent latent to improve inflammation as well as defend against cartilage disorder in OA.

## 1. Introduction

Osteoarthritis (OA) is the most common musculoskeletal disease and generally has characteristics such as joint progressive damage to articular cartilage and subchondral bone, decomposition of cartilage, and subchondral bone stiffness [1,2]. During the progression of OA, diverse changes happen including oxidative stress, inflammation, apoptosis, cartilage loss matrix, and autophagy. The pathological findings that occur in cartilage are chondrocyte apoptosis, senescence, and diminution extracellular matrix (ECM) synthesis. Pro-inflammatory cytokine expression plays important roles in the evolutionary process of OA by contributing to the degradation and erosion of cartilage [3]. These inflammatory factors, including TNF-α and IL-1β, are known as catabolic cytokines involved in the degradation of ECM by promoting the expression of MMP and a disintegrin and metalloproteinase with thrombospondin motifs (ADAMTS) [4]. The three most important foundations for the treatment of OA are pain relief, improvement of joint function, and delay of disease progression [5].

Icariin (ICA), a traditional Chinese herbal medicine, is a prenylated flavonol glycoside and is a well-known major compound derived from *Herba epimedium* [6,7,8]. It possesses a wide range of pharmacological effects such as anti-inflammatory, antioxidant, anti-cancer, and anti-atherosclerotic properties [9,10]. In addition, ICA not only promotes osteogenic differentiation but also enhances osteogenesis through various pathways, including the mitogen-activated protein kinase (MAPK) signaling pathway, phosphoinositide 3-kinase (PI3K)/protein kinase B (AKT) signaling pathway, and β-catenin signaling pathway [11,12,13]. Our previous studies reported that β-cyclodextrin/alginate (β-CD-ALG) conjugate with ICA and ICA-immobilized nanodiamonds (NDs) promoted osteogenic capacity [14,15]. Some other reports have demonstrated that ICA was a safe and powerful chondrocyte anabolic agent capable of reducing the degradation of the extracellular matrix (ECM), which protected against OA by inhibiting the overexpression of MMP-13 and suppressing pro-inflammatory mediators in chondrocytes [16,17,18]. Moreover, ICA not only reduces H_2_O_2_-stimulated human umbilical vein endothelial cell apoptosis, but also inhibits the nuclear factor kappa B (NF-κB) signaling pathway in macrophages [10,19]. Thus, ICA is attracting a lot of attention as a promising material with chondroprotective effects. 

Tannic acid (TA), a hydrolysable polyphenol compound, can functionalize the surface of various organic and inorganic materials due to its phenolic hydroxyl groups, galloyl groups, and catechol groups [20,21,22]. Our previous research confirmed osteogenic activity by delivering bone morphogenic protein-2 (BMP-2) immobilized on TA anchored on the PCL scaffold (Tannylated PCL scaffold), and previous research demonstrated that a TA-anchored E7/P15 peptide with an alginate scaffold promoted cartilage and subchondral bone regeneration in rabbit osteochondral defect model [21,22]. More recently, our groups fabricated a self-assembled TA-alendronate (ALN) nanocomplex through hydrogen bonding and confirmed that this enhanced the antioxidant, anti-inflammatory, and osteogenic effects in MC3T3-E1 cells (osteoblast-like cells) [23]. TA is desired due to its nanovehicle properties which mean it can form self-assembled cross-linked networks through the role of hydrogen bond donors, and thus can bind drug compounds primarily via hydrophobic interactions [24,25,26]. 

Multiple nanomaterials have been utilized to facilitate biomedical treatments, such as therapeutic and diagnostic platforms [27,28]. Among these nanomaterials, ND is composed of carbon-founded materials that contain graphite, graphene, carbon nanotubes, and fullerene, and is attracting attention as a promising nanomaterial due to its biocompatibility, good mechanical strength, high surface functionality, and colloidal stability [15,29,30]. The ND surface can be easily functionalized so that various drugs or proteins are linked via non-covalent and covalent bonds [31,32]. Ryu et al. [29] fabricated ALN-conjugated NDs to show induced alkaline phosphatase (ALP) activity in MC3T3-E1 cells. Cheng and colleagues [30] reported that a D-α-tocopherol polyethylene glycol 1000 succinate (TPGS)-coated ND carrier improved colloidal dispersibility and oral administration of curcumin. However, there are no previous studies on OA targets using NDs as the delivery vehicle. Therefore, the main purpose this study is to determine whether ICA-anchored NDs with laden TA (ICA/TA-NDs) can effectively treat OA.

## 2. Results and Discussion

### 2.1. Characterization of NDs with TA and/or ICA

The fabrication of ICA/TA-NDs consisted of two stages (Figure 1a): The first step was the TA coating on the NDs surface via π-π interactions. The second step was the synthesis of ICA-anchored TA-NDs. To achieve the ICA/TA-NDs, the ND surfaces were changed with TA and then anchored with ICA. The laden amount and efficiency of TA on the TA-NDs was 9.70 ± 0.29 μg and 97.04 ± 2.89%. The forms of the NDs with TA and/or ICA were investigated by scanning electron microscopy (SEM) and transmission electron microscopy (TEM) (Figure 1b,c). The sizes of individual NDs in all groups were approximately 10 nm and they were spherical in shape (Figure 1b). Through SEM coupled with energy-dispersive (SEM-EDS) X-ray spectroscopy, the main surface elements of NDs in all groups were carbon and oxygen. TEM images showed that each ND group possessed nano-sized clusters in aqueous conditions (Figure 1c). In addition, Figure 1d shows that the hydrodynamic sizes were 176.93 ± 2.12 nm for NDs, 181.40 ± 5.63 nm for TA-NDs, 184.20 ± 2.75 nm for ICA(10 μg)/TA-NDs, and 190.37 ± 3.82 nm for ICA(50 μg)/TA-NDs, respectively. Our previous research confirmed that drug-laden NDs exhibited nano-sized clusters [15]. Consistent with our previous studies, each ND group showed nano-bulk shapes. The confirmed zeta potential rates of the NDs, TA-NDs, ICA (10 μg)/TA-NDs, and ICA (50 μg)/TA-NDs were −75.9 ± 2.30 mV, −48.9 ± 1.60 mV, −68.6 ± 0.90 mV, and –73.4 ± 0.40 mV (Figure 1d), respectively.

The surface modification of NDs with TA and/or ICA was investigated by an ATR-FTIR (Figure 2). In the ATR-FTIR, TA and/or ICA-modified NDs showed major characteristic peaks at 3300–3600 cm^−1^ (O-H stretching), 2920 cm^−1^ (C-H stretching), 1652 cm^−1^ (amide I, C=O asymmetric and symmetric stretching), 1598 cm^−1^ (amide II, C-N stretching coupled with NH bending modes), and 1260 cm^−1^ (C-O vibration peak). In our previous study, the ATR-FTIR spectra of ICA-functionalized dopamine-coated NDs (DOPA-NDs) showed two major peaks, including C=O and C-N stretching peaks, indicating the presence of ICA on the ND surface [15]. After coating the NDs with ICA, these distinctive ATR-FTIR spectral peaks of ICA were also detected in the ATR-FTIR of the ICA (10 μg)/TA-NDs and ICA (50 μg)/TA-NDs. These results indicated that the ICA and TA on the ND surface was formed by the hydrogen bonding interaction.

### 2.2. ICA Release from NDs and Its Cytotoxicity 

A previous study reported that ICA (at above 100 ng/mL) showed anti-inflammatory effects [33]. Thus, we prepared two ICA/TA-NDs by loading ICA (10 and 50 μg) on TA-NDs. The loaded amounts and efficiencies of ICA per 1 mg of NDs were 7.86 ± 0.14 μg (78.61 ± 1.42%) for ICA (10 μg)/TA-NDs and 39.28 ± 0.94 μg (78.56 ± 1.88%) for ICA (50 μg)/TA-NDs. The release styles of ICA from ICA (10 μg)/TA-NDs and ICA (50 μg)/TA-NDs for 28 days are shown in Figure 3a. On day 1, the ICA (10 μg)/TA-NDs and (50 μg)/TA-NDs were 3.90 ± 0.15 μg (54.93 ± 2.13%) and 24.55 ± 0.35 μg (44.96 ± 0.83%), respectively. During the 28 day release period, the released amount and percentages of ICA were 7.02 ± 0.06 μg (98.81 ± 0.90%) for ICA (10 μg)/TA-NDs and 43.54 ± 0.95 μg (96.48 ± 2.25%) for ICA (50 μg)/TA-NDs. The TA molecule can have a strong affinity with various biomolecules through electrostatic interaction, hydrogen bonding, and hydrophobic interaction. Lee et al. [21] reported that a bone morphogenic protein-2 (BMP-2)-delivering TA-coated polycaprolactone (PCL) scaffold exhibited controlled and sustained BMP-2 release. Similarly to the previous study, we showed that ICA/TA-NDs showed controlled and sustained ICA release. 

The cytotoxic investigation of NDs with TA and/or ICA in RAW 264.7 cells was evaluated using the CCK-8 assay. The cytotoxic effect of each group on the viability of RAW 264.7 cells was studied for 48 h (Figure 3b). Cell viabilities after treatment with each group revealed more than 95% of that seen in the control group for 24 and 48 h. Thus, no sample produced any obvious cytotoxicity in the RAW 264.7 cells. 

### 2.3. Anti-Inflammatory Activity

Macrophages, which play a vital role in the development of OA, have a plastic phenotype and can differentiate into other phenotypes in response to alterations in the microenvironment [34,35]. In particular, activated macrophages can be divided into two main types, from M1 macrophages with inflammatory effects to M2 macrophages with anti-inflammatory and tissue regeneration functions [36,37]. To activate macrophages, LPS was utilized. It was described that LPS-activated cells upregulate various pro-inflammatory factors [21,23]. As shown in Figure 4a–f, compared to control cells (no LPS treatment), LPS-activated cells markedly enhanced the mRNA expressions of all pro-inflammatory mediators, whereas the mRNA levels of anti-inflammatory factors had no effect for 3 days. The mRNA levels of these pro-inflammatory mediators in cells were efficiently reduced by exposure to ICA/TA-NDs in a dose-dependent manner, whereas the mRNA levels of anti-inflammatory in cells were significantly upregulated by treating with ICA/TA-NDs in a dose-dependent manner.

In particular, ICA (50 μg)/TA-NDs not only significantly suppressed the mRNA levels of pro-inflammatory mediators in LPS-stimulated cells, but also markedly upregulated the mRNA levels of anti-inflammatory mediators in LPS-stimulated cells, compared with the other ND groups. Consistent with previous studies [10,17], ICA/TA-NDs suppressed the levels of pro-inflammatory factors as well as increased the levels of anti-inflammatory factors in LPS-activated macrophages, indicating that ICA/TA-NDs are effective systems to prohibit inflammatory response.

### 2.4. Cytokine Production in Supernatant

To further confirm the anti-inflammatory effects of ICA/TA-NDs on the cytokine levels of the supernatant in LPS-stimulated RAW 264.7 macrophages, cells were treated with DMEM containing NDs with TA and/or ICA for predesigned time periods of 24, 72, and 120 h. Compared with the control group (no treatment), the levels of IL-6 and TNF-α production in macrophages treated with LPS significantly increased in a time-dependent manner. ICA/TA-NDs significantly inhibited the production of IL-6 and TNF-α in LPS-stimulated RAW 264.7 macrophages in a dose-/time-dependent manners (Figure 5).

### 2.5. Radiological, Gross, and Histological Investigations

Both surgical and spontaneous animal models of OA require a long time to present classical features of OA, including pain and converts in joint structure [38]. Therefore, animal models that replicate the behavioral, biochemical, and pharmacological parameters of OA-related joint discomfort are of great worth. MIA, known as a metabolic inhibitor, inhibits glyceraldehyde 3-phosphate dehydrogenase activity in chondrocytes when injected into the joint, resulting in glycolysis and final cell death [39,40]. The advanced loss of chondrocytes displays histological and morphological alterations in articular cartilage which are very similar to those observed in human OA [41]. Moreover, MIA-induced OA models reveal inflammation and pain, including cartilage and subchondral erosion [39]. Therefore, the MIA-activated OA rat model was picked to assess the anti-inflammatory effect of ICA/TA-NDs.

An in vitro release study showed that ICA/TA-NDs released almost all of the ICA (over 97%) for 4 weeks. Here, it is wondered whether a single dose of ICA/TA-NDs can attenuate disease progression and cartilage degradation in OA rat models at 8 weeks. Therefore, 8 weeks after ND, ICA (10 µg)/TA-ND, and ICA (50 µg)/TA-ND treatments, micro-CT images were acquired to confirm the grade of progression of OA (Figure 6A). The MIA only and MIA + ND treatments were observed to significantly present irregular articular surface, surface erosions, subchondral bone sclerosis, and severe femoral condyle destruction. After intra-articular treatment with two ICA/TA-NDs, subchondral erosions and irregular articular surfaces were markedly decreased compared to rats receiving MIA only and the MIA + NDs group. In particular, ICA (50 µg)/TA-NDs presented little irregularity of the joint surface and the least erosion of articular cartilage. Tang et al. [42] reported that treatment of ICA ameliorated the articular cartilage and head of OA rats in a dose-dependent manner. Consistent with a previous study, ICA/TA-NDs alleviated irregularity of the joint surface and erosion of articular cartilage in a dose-dependent manner. 

Gross findings were evaluated to further assess the grade of OA progression (Figure 6B). MIA-induced and ND-treated rat cartilage surfaces confirmed several characteristic morphological alterations of cartilage surfaces, such as yellow discoloration, erosion extending to the deep layers, and even partial dissection of the subchondral bone. Treatment with ICA (10 µg)/TA-NDs in an OA joint led to less degradation of cartilage surfaces. Furthermore, the lowest degradation of the articular cartilage surfaces of rats was observed in the ICA (50 µg)/TA-ND-treated group. 

H and E staining of tissues from the MIA only and MIA + NDs groups observed surface grimness together with destruction of the cartilage and meniscus (Figure 7a). Rats exposed with ICA (10 µg)/TA-NDs also exhibited surface roughness but gentle cartilage degradation compared with the MIA only and MIA + NDs groups. The ICA (50 µg)/TA-ND-treated group displayed mild surface roughness as well as the best level of cartilage among the treated groups due to the anti-inflammatory effects of the ICA released from the ICA/TA-NDs. The outcomes of safranin-O staining displayed the rate of proteoglycan destruction of articular cartilage due to OA progression (Figure 7b). The MIA only and MIA + NDs groups observed a loss of proteoglycans to the deep area of the cartilage layers with cartilage fibrillations and erosions. The ICA (10 µg)/TA-ND group displayed a loss of proteoglycans in the middle zone of the cartilage layers, compared with the MIA only and MIA + NDs groups. However, the ICA (50 µg)/TA-ND-treated group exhibited the greatest portion of proteoglycans in the cartilage and a light loss of proteoglycans in the superficial zone of cartilage layers. Zu et al. [33] cognized that ICA-treated OA rats revealed less severe destructions and erosion of cartilage. More recently, treatment with ICA ameliorated cartilage degeneration of the cartilage layers in a dose-dependent manner [42]. These previous outcomes were consistent with our results, indicating that ICA/TA-NDs alleviated cartilage degeneration of the cartilage layers in a dose-dependent manner. OARSI grading was conducted to assess the in vivo suppressing effects of ICA/TA-NDs on OA progression in rats (Figure 7c). There were significant differences in OARSI grades after treatment with ICA/TA-NDs in comparison to MIA induction and NDs only (** *p* < 0.01). In addition, the OARSI grades from ICA (50 µg)/TA-ND injection observed the lowest worth, relative to those from the other groups (** *p* < 0.01). These results indicate that OA progression is significantly suppressed via the sustained and long-term release of ICA from ICA/TA-NDs.

### 2.6. Effect of ICA/TA-NDs on Inflammatory Factors in Serum Levels

Pro-inflammatory factors (i.e., TNF-α, IL-1β, and IL-6) and matrix-degrading enzymes (containing MMP-1, MMP-13, MMP-10, and MMP-3) are significantly promoted and assist important roles in the maintenance of chronic inflammation and tissue damage during the progression of OA [34,35,43]. The anti-inflammatory factors (i.e., IL-4, IL-10, insulin-like growth factor (IGF), and transforming growth factor-β (TGF-β)) in OA play a role in suppressing the action of pro-inflammatory cytokines, downregulating the expression level of MMPs and promoting the synthesis of chondrocyte proteoglycan and collagen type II [44,45]. For confirming the anti-inflammatory effects of ICA/TA-NDs, the serum levels of IL-6, IL-10, MMP-3, and TNF-α in the MIA-induced OA model were measured by the ELISA method at 8 weeks after treatment with ICA/TA-NDs. Compared with the normal group, the MIA group had upregulated serum levels of IL-6, MMP-3, and TNF-α, whereas the serum level of IL-10 did not elevate (Figure 8). The ICA/TA-ND treatment groups decreased the serum levels of IL-6, MMP-3, and TNF-α in a dose-dependent manner, compared with the ND treatment groups. Moreover, the ICA/TA-ND groups increased the serum level of IL-10 of the anti-inflammatory component in a dose-dependent manner. 

The ICA (50 µg)/TA-ND group not only significantly suppressed the serum levels of IL-6, MMP-3, and TNF-α, but also markedly increased the serum level of IL-10, compared to the other groups at 8 weeks. Previous studies demonstrated that ICA alleviated a rabbit OA model via suppressing the NF-κB pathway as well as decreased a rat OA model through NLRP3-mediated pyroptosis [33,46]. These previous results were coincidental with our results, indicating that ICA/TA-NDs suppressed pro-inflammatory factors in serum levels in a dose-dependent manner associated with ICA.

In vitro and in vivo findings are consonant with previous studies performed in cells and animal models. However, there are still little studies explaining the attenuation of inflammation and cartilage degradation in OA animal models using ICA. However, the ICA/TA-NDs used in this study have been displayed to be effective in improving inflammation and attenuating cartilage degradation.

## 3. Materials and Methods

### 3.1. Nanodiamonds (NDs) Modified by Tannic Acid (TA) and Icariin (ICA)

To anchor icariin (ICA, Tokyo Chemical Industry Co., Ltd., Tokyo, Japan), carboxylated nanodiamonds (NDs, particle size: <10 nm, Tokyo Chemical Industry Co., Ltd., Tokyo, Japan) were first modified by tannic acid (TA, Tokyo Chemical Industry Co., Ltd., Tokyo, Japan) as follows: To start, NDs (10 mg) were put into phosphate-buffered saline (PBS; Welgene Inc., Gyeongsan-si, Gyeongsangbuk-do, Republic of Korea) solution (pH 7.4) with dissolved TA (10 μg/mL) and then slightly shaken for 24 h under room temperature (RT). After that, TA-laden NDs were cleaned 3 times with double deionized water (DDI), followed by lyophilization for 2 days and stored in a refrigerator. The PBS solution used in TA-laden NDs was acquired to identify the loading amount of TA on NDs. Then, the amount and efficiency of TA were determined based on the standard curve for TA (Y = 0.0074x + 0.1461, R^2^ = 0.99; TA concentration range of standard curve: 0 to 500 μg/mL). TA-laden NDs are hereinafter referred to as TA-NDs. For the anchoring of ICA on the ND surface, TA-NDs (10 mg/mL) were dispersed in PBS solution with probe sonication (KFS-250 N, power: 45 W, Korea Process Technology Co., Ltd., Seoul, Republic of Korea) and ICA (10 or 50 μg/mL) was added into solution mentioned above, followed by incubation for 24 h. After incubation for 24 h, all samples were rinsed 2 times with DDI at 3000 rpm for 10 min at 4 °C using Smart R17 Centrifuge (Hanil Science Industrial, Incheon, Republic of Korea), followed by freeze-drying for 2 days. ICA (10 μg)-anchored TA-NDs and ICA (50 μg)-anchored TA-NDs were designated as ICA (10 μg)/TA-NDs and ICA (50 μg)/TA-NDs, respectively.

### 3.2. Characterization

NDs, TA-NDs, ICA (10 μg)/TA-NDs, and ICA (50 μg)/TA-NDs were scattered in ethanol (C_2_H_5_OH) using a probe sonication in ice bath for 3 min, placed drop-by-drop on a glass coverslip, and dried to remove ethanol. The shape of each group was displayed by field emission TEM (JEM-F200, JEOL Ltd., Tokyo, Japan). The morphology and surface elemental composition of each sample was exhibited by field emission SEM (FE-SEM, S-4700, Hitachi, Japan) supplemented with EDS. To analyze particle sizes, particle distributions, and surface charge, each group (100 μg) was re-suspended in DDI using an ultra-sonicator at 4 °C. The scattering of all samples was performed by Nano Particle Analyzer (SZ-100V2, HORIVA, Kyoto, Japan) instrument supplemented with a He-Ne laser based on a dynamic light scattering (DLS) technique. To assess surface modification of all groups, attenuated total reflectance–Fourier transform infrared (ATR-FTIR) spectroscopy was used. ATR-FTIR spectra were acquired by 4000 and 500 cm^−1^ at a 4 cm^−1^ speed. To quantify the drug-loading amount and efficiency, the residual ICA solution was analyzed at 290 nm by a Multimode Reader (Thermo Scientific Inc., Walthan, MA, USA). Then, the drug amount and efficiency were investigated according to the standard curve for ICA (Y = 0.011x + 0.979, R^2^ = 0.9989; ICA concentration range of standard curve: 0 to 200 μg/mL). 

### 3.3. In Vitro ICA Release from ICA/TA-NDs

The in vitro release amount of ICA from the ICA/TA-NDs was demonstrated as follows: To start, 10 mg of ICA (10 μg)/TA-NDs and ICA (50 μg)/TA-NDs were dispersed in PBS solution (pH 7.4) using a bath-type device, then were added to a dialysis bag (molecular weight cut off 6–8 kDa, Spectrum Laboratories, Inc., Rancho Dominguez, CA, USA) and transferred to a 50 mL tube containing 20 mL of PBS solution. The 50 mL tube with dialysis bag was then shaken at 100 rpm under warm conditions (37 °C). At predesignated time points, supernatant was acquired and the same amount of new PBS solution was added. The amount of ICA released was recorded at 290 nm using a Multimode Reader.

### 3.4. Cytotoxicity

To assess cytotoxic effect of each sample, RAW 264.7 macrophages (Korea Cell Line Bank, Seoul, Republic of Korea) were plated in 96-well cell culture plate at 1 × 10^4^ cells/well, followed by cultivation with DMEM (Thermo Fisher Scientific, Rockford, IL, USA) including 10% FBS (Thermo Fisher Scientific) and 1% antibiotics (Thermo Fisher Scientific) for 24 h. After cultivation for 24 h, cells were exposed to DMEM containing each sample (100 µg/mL). After incubation for 24 or 48 h, CCK-8 solution (Cell Counting Kit-8, Dojindo, Inc., Rockville, MD, USA) was added to each cell and followed by further incubation for 1 h at 37 °C in the dark. The optical density was monitored at 450 nm using a multimode reader. Cells not cultured from each sample treatment were used as a control group. All experiments per period were repeated three times. The number of samples/each group was 4. 

### 3.5. Anti-Inflammatory Effects of ICA/TA-NDs

To confirm anti-inflammatory activity of each sample, cells (1 × 10^5^ cells/well) were seeded on 24-well culture plates including DMEM, cultivated for 24 h, and then treated with 100 ng/mL of lipopolysaccharide (LPS, Sigma) in the presence or absence of each sample (100 µg/mL). After LPS treatment for 3 days, total RNA was acquired using extracted using TRizol reagent (Life technologies, New York, NY, USA) and cDNA was generated from total RNA (1 μg) using AccuPower RT PreMix. The primer sequences of the target genes were used as follows: COX-2 (F) 5′-CAG CCA TAC TAT GCC TCG GA-3′, (R) 5′-GGA TGT CTT GCT CCG CGT TC-3′; IL-6 (F) 5′-CCG TTT CTA CCT GGA GTT TG-3′, (R) 5′-GTT TGC CGA GTA GAC CTC AT-3′, MMP-3 (F) 5′-ACC TGT CCC TCC AGA ACC TG-3′, (R) 5′-AAC TTC ATA TGC GGC ATC CA-3′; TNF-α (F) 5′-CTC CCA GAA AAG CAA GCA AC-3′, (R) 5′-CGA GCA GGA ATG AGA AGA GG-3′; IL-4 (F) 5′-ACA GGA GAA GGG ACG CCA T-3′, (R) 5′-GAA GCC CTA CAG ACG AGC TCA-3′ and IL-10 (F) 5′-GGT TGC CAA GCC TTA TCG GA-3′, (R) 5′-ACC TGC TCC ACT GCC TTG CT-3′. The ABI7300 Thermal Cycler using the SYBR Green PCR master mix was utilized to conduct real-time PCR. Target genes were normalized using GAPDH. The number of samples/each group was 4.

### 3.6. Determination of Cytokine Production in Cell Supernatant

To further evaluate anti-inflammatory effects of ICA/TA-NDs, cells (1 × 10^5^ cells/well) were seeded in each well of a 24-well cell plate with DMEM, cultured for 24 h, and exposed to 100 ng/mL of LPS in the presence or absence of each sample (100 μg/mL). At pre-designed time points (24, 72, and 120 h), the supernatants were collected and stored at −80 °C for further quantification of IL-6 and TNF-α. The amount of IL-6 and TNF-α secreted in cells was analyzed using enzyme-linked immunosorbent assay (ELISA) kits (Elabscience^®^, Houston, TX, USA). The absorbance was monitored at 450 nm using a Multimode Reader.

### 3.7. Osteoarthritis (OA) Induction in Rat

The animal study protocol was approved by the Institutional Animal Care and Use Committee of Korea University College of Medicine (protocol code: KOREA-2020-0130-C1 and date of approval: 19 March 2021). To create an osteoarthritis rat model, 8-week-old rats (Sprague Dawley (SD) male rat, DooYeol Biotech, Seoul, Republic of Korea) were housed on a constant 12 h light/12 h dark cycle at 22 ± 2 °C with ad libitum standard diet pellets (DooYeol Biotech, Seoul, Republic of Korea) and water, and were allowed to adjust to the environment for 1 week before monosodium iodoacetate (MIA, Sigma) injection. Under anesthetization, the rat received a single intra-articular injection into the left knee of 50 µL of saline-dissolving MIA (10 mg/mL) using a 23 g needle. Separately, each group (1 mg) was dispersed in saline (1 mL) using a probe sonication. On the 7th day after the injection of MIA, rats in each ND group received by intra-articular administration into the left knee joints as follows: ND group, ICA (10 μg)/TA-ND and ICA (50 μg)/TA-ND group. The injection volume and needle gauge were 100 μL and 23 g needle, respectively. The real dosage of ICA was injected 78.6 ng/rat for ICA (10 µg)/TA-NDs and 392.8 ng/rat for ICA (50 µg)/TA-NDs. In vivo studies were performed in a total of 25 rats categorized into five groups (n = 5/per group) as follows: (I) Normal, (II) MIA treatment, (III) MIA + ND treatments, (IV) MIA + ICA (10 µg)/TA-ND treatments, and (V) MIA + ICA (50 µg)/TA-ND treatments. All rats were monitored the 9th week post-MIA injection and the 8th week post-NDs with TA and/or ICA treatment. 

### 3.8. In Vivo Micro-CT Imaging

At the end of week 8, all rats were euthanized with a CO_2_ overdose, dissected of knee joint, and fixed in 4% (*v*/*v*) paraformaldehyde. Each sample was imaged with a micro-CT system (SKYSCAN 1176, Bruker Co, MA, USA). Each sample was scanned with an X-ray source with setting at 60 kV with a current of 417 μA and aluminum 0.5 mm thick filter. The pixel size was a nominal resolution of 35 μm/pixel and the rotation step was 0.5°. The cross-sectional images were reconstructed using a filtered back-projection algorithm (NRecon software, version 1.6.3.3. Bruker micro-CT, Belgium). For each scan, a stack of 286 cross-sections was reconstructed at 2000  ×  1336 pixels.

### 3.9. Histological Examination

Knee joints were isolated, fixed in phosphate-buffered 4% formaldehyde, decalcified with 5% formic acid, dehydrated, and embedded in paraffin. Tissue blocks were cut into 7 μm thickness. The sectioned tissues were used to confirm the influx of inflammatory cells by Hematoxylin and Eosin (H and E) staining and to assess the degradation of glycosaminoglycan in the cartilage by safranin-O red staining. The stained tissues were used to analyze the degenerative cartilage grade and stage using the Osteoarthritis Research Society International (OARSI) Scoring System. The histopathological grade of cartilage degeneration was monitored by structural scoring (Grade 0, surface smooth and cartilage morphology healthy; Grade 1, surface unharmed; Grade 2, surface discontinuity; Grade 3, vertical fissures (clefts); Grade 4, erosin; Grade 5, denudation; Grade 6, deformation).

### 3.10. Measurement of Pro- and Anti-Inflammatory Component Levels in Serum

At the end of in vivo study, blood was harvested from tail vein to analyze anti-inflammatory ICA/TA-NDs. Harvested bloods were centrifuged at 2500 rpm for 10 min, followed by movement to new tube, and preserved at −80 °C until analysis. The levels of IL-6, IL-10, MMP-3, and TNF-α in the serum were estimated by ELISA kits. The absorbance was monitored at 450 nm using a Multimode Reader.

### 3.11. Statistical Analysis

All experimental data are presented as the means ± standard deviations (SDs). Statistical analyses were performed via one-way ANOVA with Tukey’s multiple comparison in SigmaPlot 12.0 (Systat software, Inc., San Jose, CA, USA). *p*-values less than 0.05 or less than 0.01 were considered significant.

## 4. Conclusions

In this study, ICA/TA-NDs were successfully prepared by loading ICA on the surfaces of TA-NDS. ICA/TA-NDs showed the prolonged release of ICA and they possessed anti-inflammatory effects by inhibiting the mRNA levels of markedly pro-inflammatory factors (i.e., COX-2, IL-6, MMP-3, and TNF-α), as well as increasing the mRNA levels of anti-inflammatory cytokines (i.e., IL-4 and IL-10) in LPS-activated macrophages in a dose-dependent manner. Furthermore, they exhibited in vivo anti-inflammatory characteristics by outstandingly decreasing pro-inflammatory components (i.e., IL-6, MMP-3, and TNF-α) while increasing the anti-inflammatory IL-10 in serum levels of MIA-induced OA rat models. Due to their potent anti-inflammatory properties, ICA/TA-NDs attenuated cartilage destruction and MIA-induced OA progression in rat models. Therefore, these results suggest that ICA/TA-NDs will be useful for the treatment of pain and cartilage disorder in OA patients.

## Figures and Tables

**Figure 1 ijms-24-15977-f001:**
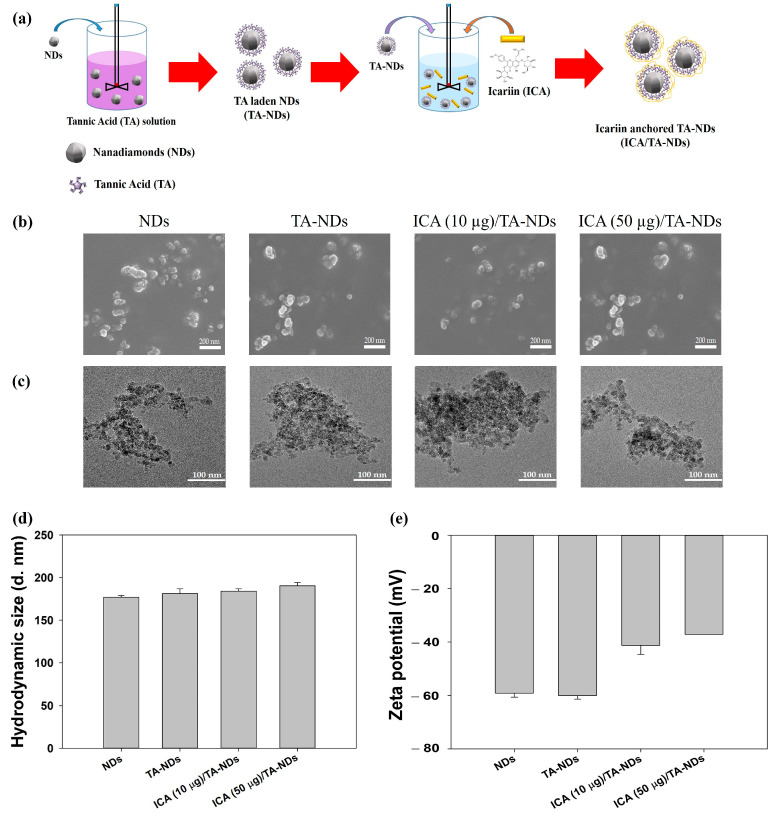
(**a**) Schematic illustration of the fabrication of ICA/TA-NDs. (**b**) Scanning electron microscopy (SEM) and (**c**) transmission electron microscopy (TEM) images of NDs, TA-NDs, ICA (10 μg)/TA-NDs, and ICA (50 μg)/TA-NDs. (**d**) Hydrodynamic sizes and (**e**) zeta potential of NDs, TA-NDs, ICA (10 μg)/TA-NDs, and ICA (50 μg)/TA-NDs.

**Figure 2 ijms-24-15977-f002:**
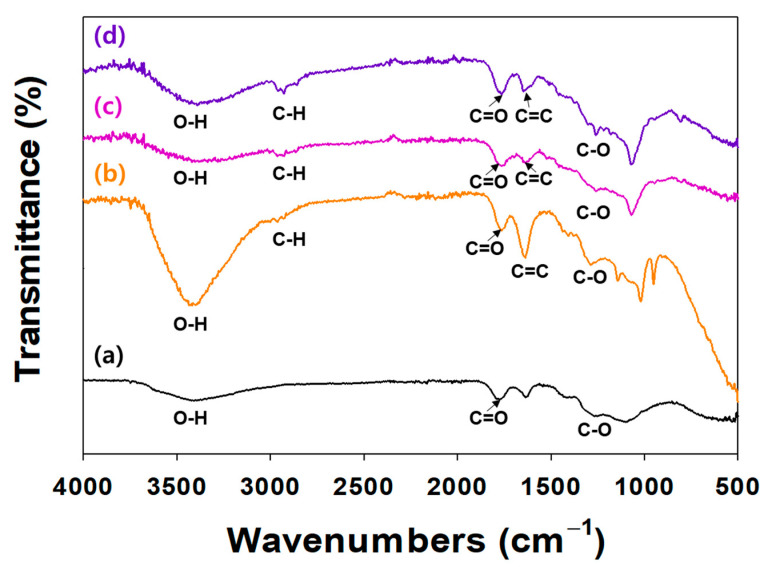
FT-IR spectra of (**a**) NDs, (**b**) TA-NDs, (**c**) ICA (10 µg)/TA-NDs, and (**d**) ICA (50 µg)/TA-NDs.

**Figure 3 ijms-24-15977-f003:**
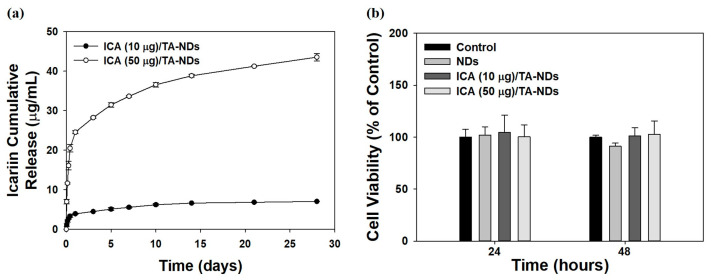
(**a**) In vitro release behavior of ICA from ICA (10 µg)/TA-NDs and ICA (50 µg)/TA-NDs for 28 days. (N = 4 per group). (**b**) In vitro cytotoxic effect of NDs, ICA (10 µg)/TA-NDs, and ICA (50 µg)/TA-NDs against RAW 264.7 macrophage cells for 24 or 48 h (N = 4 per group).

**Figure 4 ijms-24-15977-f004:**
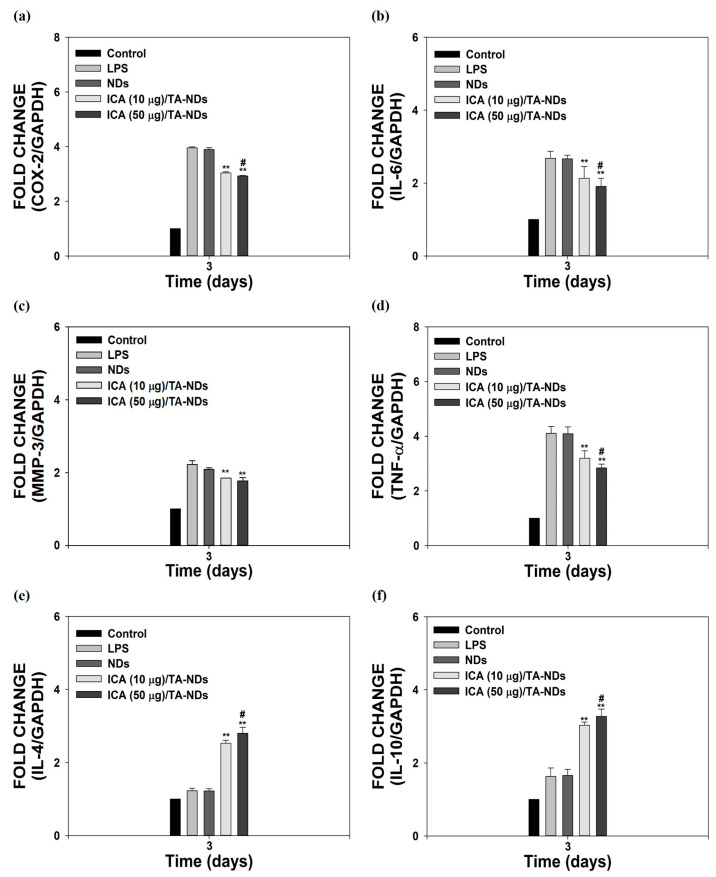
In vitro anti-inflammatory effects of ICA/TA-NDs in LPS-induced RAW 264.7 cells. mRNA levels of pro/anti-inflammatory components were assessed by real-time PCR: (**a**) COX-2, (**b**) IL-6, (**c**) MMP-3, (**d**) TNF-α, (**e**) IL-4, and (**f**) IL-10 in LPS-promoted RAW 264.7 cells 3 days after incubation with different conditions, such as NDs, ICA (10 µg)/TA-NDs, and ICA (50 µg)/TA-NDs. The significance was ** *p* < 0.001 vs. LPS-treated cells, and ^#^ *p* < 0.05 vs. ICA (10 μg)/TA-NDs. N = 4 each group.

**Figure 5 ijms-24-15977-f005:**
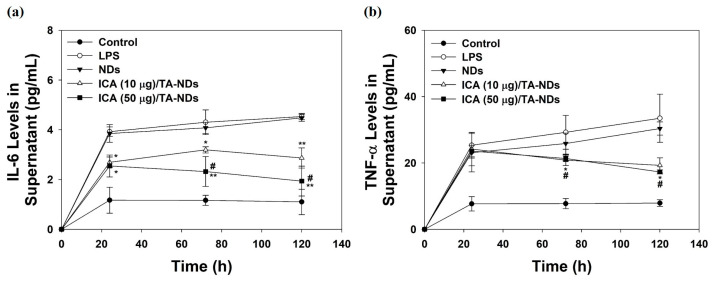
(**a**) IL-6 and (**b**) TNF-α in supernatant of cell medium secreted by LPS-activated RAW 264.7 macrophage treated with NDs, ICA (10 μg)/TA-NDs, and ICA (50 μg)/TA-NDs at 24, 72, and 120 h by ELISA assay. The significance was * *p* < 0.05 vs. LPS-treated cells, ** *p* < 0.001 vs. LPS-treated cells, and ^#^
*p* < 0.05 vs. ICA (10 μg)/TA-NDs. N = 4 each group.

**Figure 6 ijms-24-15977-f006:**
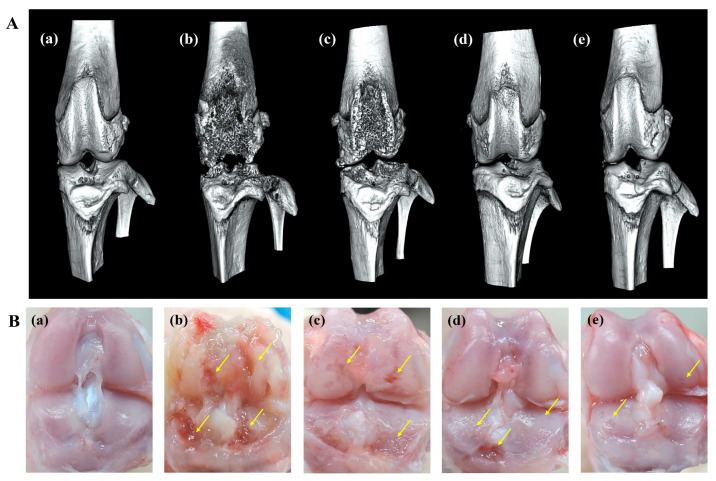
(**A**) micro-CT images and (**B**) gross finding 8 weeks after NDs, ICA (10 μg)/TA-NDs, and ICA (50 μg)/TA-NDs in MIA-induced OA rats. (**a**) Normal, (**b**) MIA only, (**c**) MIA and NDs, (**d**) MIA and ICA (10 μg)/TA-NDs, and (**e**) MIA and ICA (50 μg)/TA-NDs. Arrows represent erosions.

**Figure 7 ijms-24-15977-f007:**
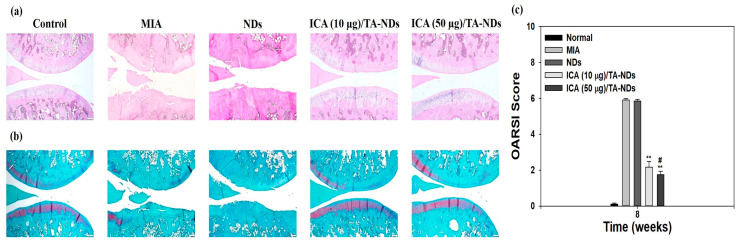
(**a**) Hematoxylin and eosin (H and E) and (**b**) safranin-O staining at 8 weeks after ND, ICA (10 μg)/TA-ND, and ICA (50 μg)/TA-ND injection in MIA-induced OA rats. (**c**) OARSI histological score graph. The significance was ** *p* < 0.001 vs. MIA-treated groups, and ^#^
*p* < 0.05 vs. ICA (10 μg)/TA-NDs.

**Figure 8 ijms-24-15977-f008:**
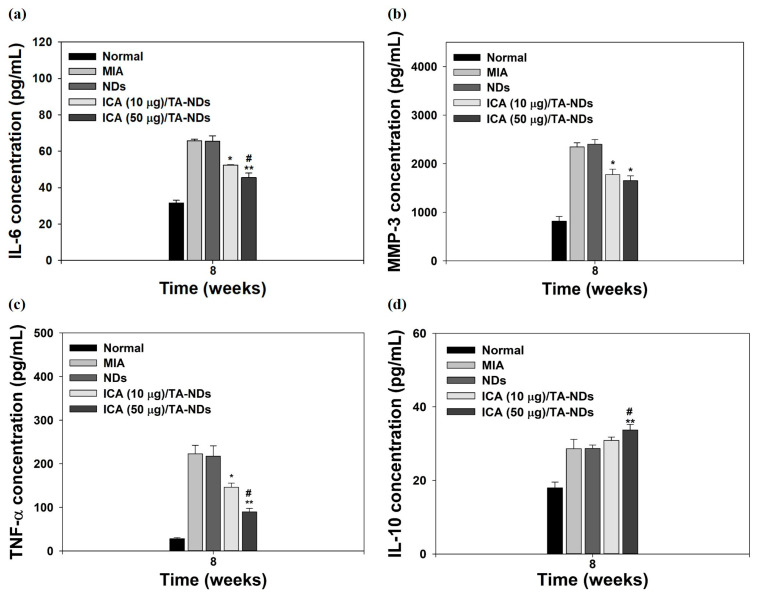
The levels of (**a**) IL-6, (**b**) MMP-3, (**c**) TNF-α, and (**d**) IL-10 at serum from MIA-induced OA rats at 8 weeks after treatment with NDs, ICA (10 μg)/TA-NDs, and ICA (50 μg)/TA-NDs. The significance was * *p* < 0.05 vs. MIA-treated groups, ** *p* < 0.001 vs. MIA-treated groups, and ^#^
*p* < 0.05 vs. ICA (10 μg)/TA-NDs.

## Data Availability

Not applicable.

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
