# Peer review of "Therapeutic Nanodiamonds Containing Icariin Ameliorate the Progression of Osteoarthritis in Rats"

_ijms, 2023, doi:10.3390/ijms242115977_

Round 1

Reviewer 1 Report

Comments and Suggestions for Authors

The authors of this manuscript explore the effect of ICA (10 μg)/TA-NDs and ICA (50 μg)/TA-NDs on treating OA. The study included in vitro and in vivo (rat model). The proposed study is commendable. The references were recent, the methods were well-written, but the discussion needs improvement. However, some drawbacks were found, as follows:

1.      In the introduction section:

·      The authors stated that "The three most important foundations for the treatment of OA are pain relief, improvement of joint function and delay of disease progression [5].," but reference number 5 was about animal models of osteoarthritis: challenges of model selection and analysis; the authors must revise this.

  1. In the results and discussion section:

·     The FTIR explanation is not sufficient; there is a missed interpretation of bands around 3500 and below 3000 cm-1, and the band positions must be assigned to the curves. In addition, the bands in the FTIR spectra are too vague to distinguish. The size of the spectra should be further adjusted for the benefit of the readers.

·    The hydrodynamic sizes of all samples have extra errors: 237.70 ± 77.0 nm for NDs, 221.90 ± 71.8 nm for TA-NDs, 216.4 ± 48.7 nm for ICA/TA-NDs, and 216.6 ± 69.7 nm for ICA/TA-NDs. The errors should be fewer than the ones mentioned in the manuscript.

·         The resolution of TEM images is very poor.

·     To thoroughly examine the nanoparticles, it is essential to include SEM and the corresponding EDS analysis in the study.

·         The correlation between pro- and anti-inflammatory markers in vitro (RAW 264.7 macrophages) and in vivo is lacking.

  1. In the materials and methods section:
  • The size of the carboxylated-nanodiamond that was purchased was not mentioned by the author.
  • Why did the author choose the doses of ICA (10 μg)/TA-NDs and ICA (50 μg)/TA-NDs?
  • Why did the author wait for the 8th week post-NDs with TA and/or ICA treatment while the release was until the 28th day? The author must explain this to the reader.
  • Did the author inject a single dose of NDs, ICA (10 μg)/TA-NDs, and ICA (50 μg)/TA-NDs? Authors must explain.

Author Response

Manuscript ID: IJMS-2672599

Reviewer-1’s Comments and Suggestions for Authors: The authors of this manuscript explore the effect of ICA (10 μg)/TA-NDs and ICA (50 μg)/TA-NDs on treating OA. The study included in vitro and in vivo (rat model). The proposed study is commendable. The references were recent, the methods were well-written, but the discussion needs improvement. However, some drawbacks were found, as follows:

In the introduction section:

Comment-1:  The authors stated that "The three most important foundations for the treatment of OA are pain relief, improvement of joint function and delay of disease progression [5].," but reference number 5 was about animal models of osteoarthritis: challenges of model selection and analysis; the authors must revise this.

Answer: Thank you for the comment. As reviewer kindly suggested, we revised the reference [5].

Correction (Line 44 ~ 46): The three most important foundations for the treatment of OA are pain relief, improvement of joint function and delay of disease progression [5].

Revised Reference: [5] Wang, H.; Ma, B. Healthcare and Scientific Treatment of Knee Osteoarthritis. J Healthc Eng 2022, 2022, 5919686, doi:10.1155/2022/5919686.

In the results and discussion section:

Comment-2:  The FT-IR explanation is not sufficient; there is a missed interpretation of bands around 3500 and below 3000 cm-1, and the band positions must be assigned to the curves. In addition, the bands in the FTIR spectra are too vague to distinguish. The size of the spectra should be further adjusted for the benefit of the readers.

Answer: As reviewer suggested, we assigned characteristic peaks, and the size of FT-IR spectra were magnified.

Correction (Line 115-116): In the ATR-FTIR, TA and/or ICA-modified NDs showed major characteristic peaks at 3300–3600 cm-1 (O-H stretching), 2920 cm-1 (C-H stretching), 1652 cm-1 (amide I, C = O asymmetric and symmetric stretching), 1598 cm-1 (amide II, C-N stretching coupled with NH bending modes), and 1,260 cm-1 (C-O vibration peak).

Revised Figure 2:

Comment-3:  The hydrodynamic sizes of all samples have extra errors: 237.70 ± 77.0 nm for NDs, 221.90 ± 71.8 nm for TA-NDs, 216.4 ± 48.7 nm for ICA/TA-NDs, and 216.6 ± 69.7 nm for ICA/TA-NDs. The errors should be fewer than the ones mentioned in the manuscript.

Answer: We made a mistake to represent the errors for the particle sizes. As reviewer indicated, we revised them in the main text.

Correction (Line 101-103): In addition, Figure 1d exhibited that the hydrodynamic sizes were 176.93 ± 2.12 nm for NDs, 181.40 ± 5.63 nm for TA-NDs, 184.20 ± 2.75 nm for ICA(10 mg)/TA-NDs, and 190.37 ± 3.82 nm for ICA(50 mg)/TA-NDs, respectively.

Comment-4:  The resolution of TEM images is very poor. Also, to thoroughly examine the nanoparticles, it is essential to include SEM and the corresponding EDS analysis in the study.

Answer: As reviewer suggested, we provided the magnified TEM images. In addition, we also include SEM images in Figure 1b and briefly described EDS results in the text.

Correction (Line 95-103): The forms of NDs with TA and/or ICA were investigated by scanning electron microscopy (SEM) and transmission electron microscopy (TEM) (Figure 1b and 1c). The sizes of individual NDs in all groups were approximately 10 nm, and they were spherical in shape (Figure 1b). Through SEM coupled with energy-dispersive (SEM-EDS) X-ray spectroscopy, the main surface elements of NDs in all groups were carbon and oxygen (data not shown). TEM images showed that each ND group possessed nano-sized clusters in aqueous condition (Figure 1c). In addition, Figure 1d exhibited that the hydrodynamic sizes were 176.93 ± 2.12 nm for NDs, 181.40 ± 5.63 nm for TA-NDs, 184.20 ± 2.75 nm for ICA(10 mg)/TA-NDs, and 190.37 ± 3.82 nm for ICA(50 mg)/TA-NDs, respectively.

Revised Figure 1:

Added sentences (Line 317-322) : NDs, TA-NDs, ICA (10 mg)/TA-NDs and ICA (50 mg)/TA-NDs were scattered in ethanol (C2H5OH) using a probe sonication in ice bath for 3 min, followed by drop-by-drop on a glass coverslip and dried to remove ethanol. The shape of each group was displayed by field-emission TEM (JEM-F200, JEOL Ltd., Japan). The morphology and surface elemental composition of each sample was exhibited by field-emission SEM (FE-SEM, S-4700, Hitachi, Japan) supplemented with EDS.

Comment-5:  The correlation between pro- and anti-inflammatory markers in vitro (RAW 264.7 macrophages) and in vivo is lacking.

Answer: To verify the correlation between pro- and anti-inflammatory markers, we analyzed IL-10 levels (as an anti-inflammatory marker) in serum using ELISA.

Corrected sentences (Line 23-25) : Animal studies exhibited that intra-articular injection of ICA/TA-NDs notably suppressed levels of IL-6, MMP-3, and TNF-α and induced level of IL-10 in serum of MIA-induced OA rat models in dose-dependent manner.

Added sentences (Line 266-270) : The anti-inflammatory factors (i.e., IL-4, IL-10, insulin-like growth factor (IGF), and transforming growth factor-β (TGF-β)) in OA play a role in suppressing the action of pro-inflammatory cytokines, downregulating the expression level of MMPs, and promoting the synthesis of chondrocyte proteoglycan and collagen type II [44,45].

Corrected sentences (Line 272-274) : Compared with the normal group, the MIA group had upregulated serum levels of IL-6, MMP-3, and TNF-α, whereas serum level of IL-10 did not elevate.

Added sentences (Line 276-277) : Moreover, ICA/TA-NDs groups increased the serum level of IL-10 of the anti-inflammatory component in a dose-dependent manner.

Corrected sentences (Line 278-280) : Especially, the ICA (50 µg)/TA-NDs group not only significantly suppressed the serum levels of IL-6, MMP-3, and TNF-α but also markedly increased the serum level of IL-10, compared to the other groups at 8th weeks.

Corrected title (Line 418) : 3.10. Measurement of pro – and anti-inflammatory component levels in serum

Added words (Line 422) : IL-10

Corrected Figure 8

Figure 8. The levels of (a) IL-6, (b) MMP-3, (c) TNF-α, and (d) IL-10 at serum from MIA-induced OA rats at 8th weeks after treatment with NDs, ICA (10 mg)/TA-NDs, and ICA (50 mg)/TA-NDs. The significance was *P < 0.05 vs LPS-treated cells, **P < 0.001 vs LPS- treated cells and #P < 0.05 vs ICA (10 mg)/TA-NDs.

Added references  

  1. Ho, Y.J.; Lu, J.W.; Ho, L.J.; Lai, J.H.; Huang, H.S.; Lee, C.C.; Lin, T.Y.; Lien, S.B.; Lin, L.C.; Chen, L.W.; et al. Anti-inflammatory and anti-osteoarthritis effects of Cm-02 and Ck-02. Biochem Biophys Res Commun 2019, 517, 155-163, doi:10.1016/j.bbrc.2019.07.036.
  2. Liu, S.; Deng, Z.; Chen, K.; Jian, S.; Zhou, F.; Yang, Y.; Fu, Z.; Xie, H.; Xiong, J.; Zhu, W. Cartilage tissue engineering: From proinflammatory and anti‑inflammatory cytokines to osteoarthritis treatments (Review). Mol Med Rep 2022, 25, doi:10.3892/mmr.2022.12615.

In the materials and methods section

Comment-6:  The size of the carboxylated-nanodiamond that was purchased was not mentioned by the author.

Answer: We added the size of commercial carboxylated-nanodiamond in the text.

Added size information (Line 298) : particle size : < 10 nm

Comment-7:  Why did the author choose the doses of ICA (10 μg)/TA-NDs and ICA (50 μg)/TA-NDs?

Answer: Previous study reported that ICA (at above 100 ng/mL) showed anti-inflammatory effects [33]. Thus, we prepared two ICA/TA-NDs by loading ICA (10 and 50 mg) on TA-NDs, and the prepared two NDs were named as ICA (10 μg)/TA-NDs and ICA (50 μg)/TA-NDs.

Added sentences (Line 128-130): Previous study reported that ICA (at above 100 ng/mL) showed anti-inflammatory effects [33]. Thus, we prepared two ICA/TA-NDs by loading ICA (10 and 50 mg) on TA-NDs.

Added reference: [33]   Zu, Y.; Mu, Y.; Li, Q.; Zhang, S.T.; Yan, H.J. Icariin alleviates osteoarthritis by inhibiting NLRP3-mediated pyroptosis. J Orthop Surg Res 2019, 14, 307, doi:10.1186/s13018-019-1307-6.

Comment-8:  Why did the author wait for the 8th week post-NDs with TA and/or ICA treatment while the release was until the 28th day? The author must explain this to the reader. Did the author inject a single dose of NDs, ICA (10 μg)/TA-NDs, and ICA (50 μg)/TA-NDs? Authors must explain.

Answer: As reviewer indicated, two ICA/TA-NDs released almost all of ICA (over 97%) for 4 weeks. However, we are wondering whether single dose of ICA/TA-NDs can attenuate disease progression and cartilage degradation in OA rat models at 8 weeks.

Added sentences (Line 206-208): In vitro release study showed that ICA/TA-NDs released almost all of ICA (over 97%) for 4 weeks. Here, it is wondering whether the single dose of ICA/TA-NDs can attenuate disease progression and cartilage degradation in OA rat models at 8 weeks.      

Reviewer 2 Report

Comments and Suggestions for Authors

In the manuscript “Therapeutic nanodiamonds containing icariin ameliorate the progression of osteoarthritis in rats” by Yu and colleagues, the authors focused on the advantages of icariin (ICA)/tannic acid (TA)-nanodiamonds on inflammation and OA progression in in vitro and in vivo models. The work is well written, the results are presented clearly and sufficiently discussed.  However, some issues need to be improved before the paper is published.

1.       Throughout the paper the authors described a dose-dependent therapeutic effect in the presence of ICA/TA-NDs. Are the differences observed in ICA (10 μg) or (50 μg)/TA-NDs treated cells statistically significant to each other? (Fig 4, 5 and 7).

The authors indicate * or ** in treated samples as statistical significance without specifying with respect to what. Please revise your statistical analysis in all the experiments and specify the statistical significance in the figure captions.

2.       Line 138. Did you mean 24 and 48 h, as the figure 3B shows?

3.       “As shown in Figure 4a-f, compared to control cells (no LPS treatment), LPS-activated cells markedly enhanced the mRNA expressions of all pro-inflammatory mediators as well as decreased anti-inflammatory factors evaluated for 3 days”. Lines 152-154.  The authors state that LPS increases the mRNA expression of pro-inflammatory markers and decreases that of anti-inflammatory ones. Actually, LPS has no effect on anti-inflammatory markers (Fig 4 e-f), please revise the sentence.

4.       (Fig. 4) The writings on the figures are very small and difficult to read. Can the figures be magnified and the quality improved to make them more readable to the reader?

5.       Please explain what the yellow arrows indicate in figure 6B. What do the authors want to highlight?

6.       Lines 257-260. In this paragraph, the authors discuss the in vivo anti-inflammatory effect of their nanosystem. Although inflammation and tissue degeneration in OA are closely related, it is suggested to move this sentence to the previous paragraph.

7.       The conclusions in this format seem to take up the abstract without providing further information, input of discussion or future perspectives. Authors should consider revising and rephrasing this paragraph.

Comments on the Quality of English Language

Minor editing of English language required

Author Response

Manuscript ID: IJMS-2672599

Reviewer-2’s Comments and Suggestions for Authors

In the manuscript “Therapeutic nanodiamonds containing icariin ameliorate the progression of osteoarthritis in rats” by Yu and colleagues, the authors focused on the advantages of icariin (ICA)/tannic acid (TA)-nanodiamonds on inflammation and OA progression in in vitro and in vivo models. The work is well written, the results are presented clearly and sufficiently discussed.  However, some issues need to be improved before the paper is published.

We appreciate the Reviewer-2 for careful reading of our manuscript and kind compliments on our work. We also thank for providing the comments that improve the clarity of the revised manuscript.

Comment-1: Throughout the paper the authors described a dose-dependent therapeutic effect in the presence of ICA/TA-NDs. Are the differences observed in ICA (10 μg) or (50 μg)/TA-NDs treated cells statistically significant to each other? (Fig 4, 5 and 7). The authors indicate * or ** in treated samples as statistical significance without specifying with respect to what. Please revise your statistical analysis in all the experiments and specify the statistical significance in the figure captions.

Answer: Thank you for the comment. As reviewer indicated, significant differences were observed between ICA (10 mg)/TA-NDs and ICA (50 mg)/TA-NDs in treated cells. We revised statistical analysis in all the experiments. The significance was #P < 0.05 vs ICA (10 mg)/TA-NDs in the figure 4, 5, 7c and 8 captions.

Answer: Thank for comments. As reviewer suggested, we revised statistical analysis in all the experiments. The significance was *P < 0.05 vs LPS, **P < 0.001 vs LPS, and #P < 0.05 vs ICA (10 mg)/TA-NDs in the figure 4, 5, 7c and 8 captions.

Correction Figure Captions 4 (171-173): The significance was *P < 0.05 vs LPS-treated cells, **P < 0.001 vs LPS-treated cells and #P < 0.05 vs ICA (10 mg)/TA-NDs.

Correction Figure Captions 5 (192-193): The significance was *P < 0.05 vs LPS-treated cells, **P < 0.001 vs LPS-treated cells and #P < 0.05 vs ICA (10 mg)/TA-NDs.

Correction Figure Captions 7 (260-261): The significance was *P < 0.05 vs LPS-treated cells, **P < 0.001 vs LPS-treated cells and #P < 0.05 vs ICA (10 mg)/TA-NDs.

Correction Figure Captions 8 (288-289): The significance was *P < 0.05 vs LPS-treated cells, **P < 0.001 vs LPS-treated cells and #P < 0.05 vs ICA (10 mg)/TA-NDs.

Comment-2: Line 138. Did you mean 24 and 48 h, as the figure 3B shows?

Answer: As reviewer suggested, we revised the sentence (Line 151)

Correction (Line 150-151): Cell viabilities after treatment with each group revealed more than 95% of that in the control group for 24 and 48 h.

Comment-3: “As shown in Figure 4a-f, compared to control cells (no LPS treatment), LPS-activated cells markedly enhanced the mRNA expressions of all pro-inflammatory mediators as well as decreased anti-inflammatory factors evaluated for 3 days”. Lines 152-154. The authors state that LPS increases the mRNA expression of pro-inflammatory markers and decreases that of anti-inflammatory ones. Actually, LPS has no effect on anti-inflammatory markers (Fig 4 e-f), please revise the sentence.

Answer: Thank you for the comment. As reviewer indicated, we revised the sentence.

Correction (Lines 160-162) : As shown in Figure 4a-f, compared to control cells (no LPS treatment), LPS-activated cells markedly enhanced the mRNA expressions of all pro-inflammatory mediators, whereas the mRNA levels of anti-inflammatory factors had no effect for 3 days.

Comment-4: (Fig. 4) The writings on the figures are very small and difficult to read. Can the figures be magnified and the quality improved to make them more readable to the reader?

Answer: Thank you for the comment. As reviewer suggested, we magnified the figure3, 4, 5, 7c and 8. Also, we increased the font size of the captions in these Figures. See the reformatted figure 4. See the revised these Figure.

Comment-5: Please explain what the yellow arrows indicate in figure 6B. What do the authors want to highlight?

Answer: Thank you for the comment. As reviewer indicated, we revised about yellow arrows in figure 6B caption. Arrows represent erosions. It was to be emphasized that the lowest degradation of articular cartilage surfaces of rats in the ICA (50 µg)/TA-NDs-treated group.

Correction (figure 6B caption) : (B) gross finding 8 weeks after NDs, ICA (10 mg)/TA-NDs, and ICA (50 mg)/TA-NDs in MIA-induced OA rats. (a) Normal, (b) MIA only, (c) MIA and NDs, (d) MIA and ICA (10 mg)/TA-NDs, and (e) MIA and ICA (50 mg)/TA-NDs. Arrows represent erosions.

Comment-6: Lines 257-260. In this paragraph, the authors discuss the in vivo anti-inflammatory effect of their nanosystem. Although inflammation and tissue degeneration in OA are closely related, it is suggested to move this sentence to the previous paragraph.

Answer: Thank you very much for the valuable comment. While moving the sentences (Lines 257-260) in previous paragraph, we made a mistake to repeat the same sentences (Lines 212-216). Therefore, we remove this sentence in the manuscript.

Comment-7: The conclusions in this format seem to take up the abstract without providing further information, input of discussion or future perspectives. Authors should consider revising and rephrasing this paragraph.

Answer: As reviewer commented, we revised conclusion part.

Correction (Line 430-441): In this study, ICA/TA-NDs were successfully prepared by loading of ICA on the surfaces of TA-NDS. ICA/TA-NDs showed the prolonged release of ICA, and they possessed anti-inflammatory effects by inhibiting the mRNA levels of markedly pro-inflammatory factors (i.e., COX-2, IL-6, MMP-3, and TNF-α) as well as increasing the mRNA levels of anti-inflammatory cytokines (i.e., IL-4 and IL-10) in LPS-activated macrophages in dose-dependent manner. Furthermore, they exhibited the in vivo anti-inflammatory characteristics by outstandingly decreasing pro-inflammatory components (i.e., IL-6, MMP-3, and TNF-a) while increasing anti-inflammatory IL-10 in serum levels of MIA-induced OA rat model. Due to their potent anti-inflammatory properties, ICA/TA-NDs attenuated cartilage destruction and the MIA-induced OA progression in rat models. Therefore, these results suggest that ICA/TA-NDs will be useful for the treatment of the pain and cartilage disorder in OA patients.

Round 2

Reviewer 2 Report

Comments and Suggestions for Authors

I recommend the publication of the paper in the present form.

Comments on the Quality of English Language

Minor editing of English language required.